

# Impact of crop residue management on crop production and soil chemistry after seven years of crop rotation in temperate climate, loamy soils

Marie-Pierre Hiel[1], Sophie Barbieux[1], Jérôme Pierreux[2], Claire Olivier[3], Guillaume Lobet[4,5], Christian Roisin[3], Sarah Garré[1], Gilles Colinet[6], Bernard Bodson[2] and Benjamin Dumont[2]

[1] TERRA Research & Teaching Center—AgricultureIsLife, Gembloux Agro-Bio Tech, Université de Liège, Liège, Belgium
[2] AGROBIOCHEM, Gembloux Agro-Bio Tech, Université de Liège, Liège, Belgium
[3] Unit Soil Fertility and Water Protection, Department of Agriculture and Natural Environment, Walloon Agricultural Research Center, Gembloux, Belgium
[4] Earth and Life Institute, Université catholique de Louvain, Louvain-la-Neuve, Belgium
[5] Agrosphere, IBG3, Forschungszentrum Juelich, Juelich, Germany
[6] BIOSE, Gembloux Agro-Bio Tech, Université de Liège, Liège, Belgium

Corresponding author
Benjamin Dumont,
benjamin.dumont@uliege.be

## ABSTRACT

Society is increasingly demanding a more sustainable management of agro-ecosystems in a context of climate change and an ever growing global population. The fate of crop residues is one of the important management aspects under debate, since it represents an unneglectable quantity of organic matter which can be kept in or removed from the agro-ecosystem. The topic of residue management is not new, but the need for global conclusion on the impact of crop residue management on the agro-ecosystem linked to local pedo-climatic conditions has become apparent with an increasing amount of studies showing a diversity of conclusions. This study specifically focusses on temperate climate and loamy soil using a seven-year data set. Between 2008 and 2016, we compared four contrasting residue management strategies differing in the amount of crop residues returned to the soil (incorporation vs. exportation of residues) and in the type of tillage (reduced tillage (10 cm depth) vs. conventional tillage (ploughing at 25 cm depth)) in a field experiment. We assessed the impact of the crop residue management on crop production (three crops—winter wheat, faba bean and maize—cultivated over six cropping seasons), soil organic carbon content, nitrate ($NO_3^-$), phosphorus (P) and potassium (K) soil content and uptake by the crops. The main differences came primarily from the tillage practice and less from the restitution or removal of residues. All years and crops combined, conventional tillage resulted in a yield advantage of 3.4% as compared to reduced tillage, which can be partly explained by a lower germination rate observed under reduced tillage, especially during drier years. On average, only small differences were observed for total organic carbon (TOC) content of the soil, but reduced tillage resulted in a very clear stratification of TOC and also of P and K content as compared to conventional tillage. We observed no effect of residue management on the $NO_3^-$ content, since the effect of fertilization dominated the effect of residue management. To confirm the results and enhance early tendencies, we believe that the

experiment should be followed up in the future to observe whether more consistent changes in the whole agro-ecosystem functioning are present on the long term when managing residues with contrasted strategies.

## INTRODUCTION

Once a crop is harvested, farmers have to decide what to do with the remaining crop residue (the above ground biomass that is cut but not harvested). Residues can be either exported and valorised as co-products (e.g., animal fodder, biogas production), or restored to the soil as such or after being burnt. Returning straw directly to the field has been promoted as a source of organic matter and a way to increase soil water holding capacity and its overall quality. As such, it is thought to help maintain, or even to some extent restore, soil fertility (*Lal et al., 2004*). If the residues are returned to the soil, farmers have to choose how to manage them using either conventional tillage or alternatives such as reduced tillage. We define conventional tillage as a tillage based on mouldboard ploughing which is commonly used in temperate regions and reduced tillage as a tillage with reduced intensity and/or depth (*Hiel et al., 2016*; the practical implementation of these techniques are specified in Table S1).

The precise impact of the restitution (or not) of residues and of the choice of tillage system to apply to the soil-plant system remains unclear and seems to be highly dependent on the pedo-climatic conditions (soil structure, moisture, macro fauna, etc.) (*Powlson et al., 2011*). For instance, soil organic carbon (SOC) generally seems to slightly increase if residues are returned to the soil, particularly in the long term (*Chenu et al., 2014*; *Autret et al., 2016*; *Merante et al., 2017*). However, the actual quantification of straw incorporation effect on soil organic carbon stocks shows conflicting results, as synthetized by *Poeplau et al. (2015)*, with studies reporting SOC losses, SOC stabilization or even non-significant or negligible impact. The effect of tillage on SOC content is less clear. While some studies show an increase of SOC with reduced or no-tillage (*Arrouays et al., 2002*; *Smith, 2007*; *Garcia-Franco et al., 2015*), others report no effect (*Dick, 1983*; *Dolan et al., 2006*; *Dikgwatlhe et al., 2014*).

As *Hiel et al. (2016)* show in their review, the impact of crop residue management on crop performance is also contradictory in the existing literature. The presence of residues seems to be detrimental to crop germination as they can form a physical obstacle for seedlings (*Arvidsson, Etana & Rydberg, 2014*), can create a cold and humid micro-climate around the seed (*Soane et al., 2012*) and provide a favourable habitat for slugs (*Christian & Miller, 1986*) and plant pathogens (*Arvidsson, Etana & Rydberg, 2014*). In general, the literature show that weather conditions are the main factor influencing crop yields (*Linden, Clapp & Dowdy, 2000*; *Dam et al., 2005*; *Soon & Lupwayi, 2012*), and sometimes an interacting explanatory factor is the residue management (*Riley, 2014*). Residue retention tends to

induce lower yields under wet weather conditions (effect on diseases and pests) (*Riley, 2014*) and higher yields in dry conditions (effect on water retention capacity) (*Linden, Clapp & Dowdy, 2000*; *Riley, 2014*). There are also several studies reporting no effect on crop yields (*Dam et al., 2005*; *Soon & Lupwayi, 2012*; *Riley, 2014*; *Brennan et al., 2014*). Some specific results show that it is important to have the information on the entire management type (i.e., residue in or out, type of tillage, tillage depth and timing, . . . ) in order to be able to assess the impact of the management on crop performance. *Van den Putte et al. (2010)* showed for example that residue retention of winter cereals and maize, combined with reduced tillage reduces yields in Europe. On the other hand, *Blanco-Canqui & Lal (2007)* have shown that residue removal can impede crop yield.

The literature on the effect of residue management on nitrogen (N) or phosphorus (P) uptake by plant is equally dispersed with no (for N: *Brennan et al., 2014*); positive (for N: *Malhi et al., 2011*; for P: *Noack et al., 2014*) or negative (for N: *Soon & Lupwayi, 2012*; for P: *Damon et al., 2014*) effects reported by different authors. These differences are generally attributed to differences in soil texture and/or initial nutrient status or residue quality (*Kumar & Goh, 1999*; *Chen et al., 2014*).

Interactions between crop residue management and the soil-water-plant system are complex and inherently depend on the pedo-climatic conditions. Local assessment and system approach are therefore necessary to come to relevant guidelines for residue management under specific pedo-climatic conditions. The objective of our study was therefore to determine the effects of contrasting crop residue management strategies on crop production and components of the soil fertility, over a period of several years. Regarding crop production, we studied how residues management strategy impacts on germination rate, biomass production and yield elaboration, along with N, P and K exportation. The soil fertility components that were dynamically followed were SOC, N, P and K contents and their repartition within the soil profile. The experiment was conducted in the loam belt under temperate climatic conditions, taking into account common crop rotations and local farming practices.

## MATERIALS AND METHODS

### Site description

The field experiment (50°33′49.6″N, 4°42′45.0″E) was established on 1.7 ha of the experimental farm of Gembloux Agro-Bio Tech, University of Liège, Belgium in 2008 and yield measurements started in 2010. The soil is a Cutanic Luvisol (*IUSS Working Group WRB, 2014*). According to the Walloon soil map (*WalOnMap, 2018*), the soil was silty with favourable natural drainage, containing 70–80% of silt, clay content of 18–22% of clay and 5–10% of sand. A characterization of the spatial variability of certain chemical parameters was carried out in 2011 (maps available in Fig. S1). Descriptive statistics is presented in Table 1.

The climate is temperate (Cfb in Köppen–Geiger classification (*Peel, Finlayson & McMahon, 2007*) with 819 mm average annual rain and 9.8 °C annual average temperature. Weather data were measured in a federal weather station located in Ernage (Belgium's Royal

**Table 1  Descriptive statistics of soil fertility indicators within the experimental fields on the 0–30 cm soil layer.**

| N[a] = 107 | pH KCl | TOC [g/kg] | P [g/kg] | K [g/kg] | Ca [g/kg] |
|---|---|---|---|---|---|
| Mean ± sd[b] | 6.79 ± 0.19 | 12.7 ± 1.2 | 0.149 ± 0.049 | 0.162 ± 0.027 | 2.56 ± 0.37 |
| CV[c] | 2.8 | 9.4 | 0.329 | 0.167 | 0.144 |
| Min/Max | 6.4/7.3 | 9.4/16 | 0.65/0.248 | 0.105/0.222 | 2.05/3.69 |

**Notes.**
[a] number of sample
[b] standard deviation
[c] coefficient of variation

**Table 2  Comparison of conventional and reduced tillage treatments.**

| Period | Conventional tillage | Reduced tillage |
|---|---|---|
| After harvest | **Stubble breaking**<br>Tool: tine stubble cultivator<br>Depth: 7–10 cm | |
| Few days before sowing | **Ploughing**<br>Tool: moldboard plough<br>Depth: 25 cm | No ploughing |
| Sowing day | **Seedbed preparation**<br>Tool: dual cultivator with tines and rolls in front of the tractor and rotary harrow followed by wedge ring roller.<br>The sowing machine is either mounted behind the wedge ring roller (cereals and faba bean) or either an extra passage with a precision spaced planter (maize) is done.<br>Depth: 7–10 cm | |

Meteorological Institute), at 2.4 km from field site. An overview of monthly temperature and rainfall during the experimental period is shown in Figs. S2 and S3.

## Experimental design and treatments

The field was designed as a Latin square disposal with four replications. Each plot was 15 m wide and 40 m long. Crop residue management was defined as the combination of two practices: (i) the fate of the crop residue and (ii) the type of tillage. Firstly the residue fate can be restitution (IN) or exportation (OUT). It has to be noted that stubble and chaff are always left on the fields, even if the rest of the residue is exported. Secondly, we considered two tillage types (see Table 2): conventional (CT, 25 cm depth) or reduced (RT, 7–10 cm depth). The different combinations of these two aspects of residue management resulted in four treatments: CT-IN, CT-OUT, RT-IN, RT-OUT.

The crop rotation during the experiment was: rapeseed (*Brassica napus*) in 2008–09, three consecutive years of winter wheat (*Triticum aestivum*) in 2009–10, 2010–11 and 2011–12, mustard (*Sinapis alba*) cover crop in 2012–13, faba bean (*Vicia Faba*) in 2013, winter wheat in 2014, oats (*Avena sativa*) and peas (*Pisum sativum*) mixed as cover crop in 2014–15, and finally maize (*Zea mays*) in 2015. Sowing densities were 300 kernels/m$^2$ for winter wheat, 50 kernels/m$^2$ for faba bean, and 13 kernels/m$^2$ for maize. Sowing process is detailed for each crop in Table S1.

**Table 3  Details of crop specific measurement protocols.**

|  | Winter wheat | Faba bean | Maize |
|---|---|---|---|
| Germination rate | Four repetitions of a square of 0.25 m$^2$ | Four repetitions of a square of 0.25 m$^2$ | Two plant rows of 10 m long |
| Above-ground biomass | Two repetitions of 3 plant rows of 50 cm long | Two repetitions of a square of 0.25 m$^2$ | Two plant rows of 3 m long |
| Harvest | With experimental combine of 2 m wide | With experimental combine of 2 m wide | With experimental combine of 1.5 m wide i.e., 2 sowing lines |

N fertilisation (liquid N, UAN at 39%) followed the regional standards depending on the type of crop. Rapeseed received two applications (at stem elongation stage: 31 and 32–50 on BBCH scale (*Meier et al., 2009*) with a total of 160 kg of N/ha. Three applications were provided to winter wheat (at tillering, stem elongation and flag leaf stage (26, 30 and 37–39 on BBCH scale)) with a total of 180 kg of N/ha. Faba bean was not fertilised and maize crop was fertilised with 120 kg of N/ha before sowing. There was no external addition of P or K. Crop protection measures corresponded to the regional standards.

The detailed crop protocols (crop management, crop harvest, residue exportation, soil tillage, fertilization and crop protection treatments) are available in Table S1.

## Crop sampling and analyses

We monitored the germination rate and growth dynamics during the season with an adapted protocol for each crop type (Table 3). The determination of the germination rate consisted in counting the number of seedlings on a definite area (Table 3). To quantify above-ground biomass, plants were collected (according to crop protocol in Table 3) and their different parts (shoot and ears, pods or cobs) were separated, counted and oven-dried at 60 °C for 72 h. Grain yield was assessed with an experimental harvester adapted to each crop by one passage per plot (40 m long on a width dependent on the harvester, see specific crop protocol in Table 3). To quantify the amount of remaining crop residues on the field, residues (i.e., OUT plots: stubble and chaff, IN plots: all residue) were collected over a surface of 0.5 m wide and 2 m long immediately after harvest, dried, weighed. These samples were also used to quantify the NPK content of the remaining crop residues. Composite grain samples (maize and wheat grains; faba been seeds) of 1 kg were prepared from the harvest hopper (one sample per plot) for quantification of NPK grain content. Both grains and residues were crushed before analysis. N content was measured using the Kjeldahl method (*Bradstreet, 1965*). Phosphate and potassium (K) levels in plants were measured using a modified protocol of *Zasoski & Burau (1977)*. Samples were first treated by a concentrated acid mix of $HNO_3$ and $HClO_4$ (1:1) (15 ml per g of sample). K content was measured by a flame atomic absorption spectrometric method (Spectrometer Varian 220). P was measured by colourimetry with molybdate and ammonium vanadate at 430 nm (Nanocolor UV/VIS; Macherey-Nagel, Duren, Germany). NPK content (kg/ha) were calculated by multiplying the nutrient content (%) by the biomass of the residue or grain (kg/ha).

## Soil sampling and analyses

Twice a year around April and October, we took ten soil subsamples (with a gouge auger of 2 cm diameter) to form a composite sample per plot at 0–10 cm, 10–20 cm and 20–30 cm depth. The fall sampling was usually either made after spring crop harvest and before winter wheat sowing or after winter wheat harvest and cover crop sowing. The spring sampling was made when climatic conditions were again favourable for winter wheat growth or after spring crop sowing. TOC was determined on a 1g of dry soil (ground at 200 $\mu$m) by the Walkley-Black method (*Blakemore, 1972*): oxidation with $K_2Cr_2O_7$ and $H_2SO_4$; titration of the excess of $K_2Cr_2O_7$ with Mohr Salt (($NH_4$)$_2$Fe(SO$_4$)$_2 \cdot 6H_2O$). Available soil nutrients were measured by stirring a 10 g sample of soil (air-dried and sieved at 2 mm) for 30 min in 50 ml of solution ($C_2H_7NO_2$ 0.5 M and EDTA 0.02 M at pH 4.65 (*Lakanen & Erviö, 1971*)). After filtration, for the cations measurement, atomic emission was used for K while P was determined by colourimetry (colour reaction of *Murphy & Riley, 1962*, Nanocolor UV/VIS; Macherey-Nagel, Duren, Germany).

In addition to the two overall soil sampling campaigns per growing season, soil nitrate content was measure more frequently to catch the dynamic of uptake during the growing phase of the main crops. Composite humid soil samples based on eight subsamples (sampled with gouge auger of 2 cm diameter) were used per plot at three depths: 0–30 cm, 30–60 cm and 60–90 cm. We used KCl extraction and a colourimetry method of reduction of nitrate to nitrite (using Cadmium or Hydrazine) with a determination of nitrite ions by the modified Griess-Ilosvay reaction (*Bremner, 1965*; *Guiot, Goffart & Destain, 1992*).

## Statistical analyses

Statistical analyses were performed with R software (*R Core Team, 2015*). The statistical analyses were systematically applied to assess the effect of crop residue management on crop and soil measurements, as follows. First, a 2-way ANOVA was performed, including the soil tillage and residue fate as fixed factors (with interaction) and the plot position (line and columns of the Latin square) as random factors. In case no interaction was highlighted between the fixed factors, we compared on the one hand, IN *versus* OUT treatments, and, on the other hand, RT *versus* CT treatments. These comparisons were then immediately made on the basis of the results of the 2-way ANOVA test. Contrarily, when an interaction between the fixed factors was significant, the four treatments (CT-IN, CT-OUT, RT-IN and RT-OUT) were intercompared and ranked using a post-hoc test (Student-Newman-Keuls—SNK). Analyses of variance (2-way ANOVA) and SNK tests were performed with the *agricolae* package (*Mendiburu & Simon, 2015*). The conditions of application of the ANOVA test (normality of the distribution and homoscedasticity) were systematically checked on the residuals of the ANOVA, using respectively a Shapiro–Wilk test and a Bartlett test.

To study the evolution of soil parameters over the years, a linear mixed effects model was fitted using the *lme4* package (*Bates et al., 2014*). To evaluate possible difference between treatments on the entire profile or per depth, the model was used with soil tillage and residue fate and their interaction as fixed factors, while dates and plots were random effects. To estimate whether stratification occurred in the soil parameters per crop residue

**Table 4** **Descriptive statistics of crop residues dry weight remaining on the field (%, ±standard error) for the crops grown during the study period 2009–2015.** For each crop, treatments means with different letters are significantly different (ANOVA, $p$-value $< 0.05$).

| | Interaction between fixed factors | | | | No interaction between factors | | | |
| | Crop residue management | | | | Residue fate | | Tillag type | |
| | CT-IN | CT-OUT | RT-IN | RT-OUT | IN | OUT | CT | RT |
|---|---|---|---|---|---|---|---|---|
| Rapeseed 2008–09 | $8.17 \pm 0.35$ | $1.54 \pm 0.08$ | $8.78 \pm 0.38$ | $1.68 \pm 0.08$ | $8.48^a \pm 0.26$ | $1.61^b \pm 0.06$ | $4.86^a \pm 1.26$ | $5.23^a \pm 1.35$ |
| WW 2009–10 | $4.80 \pm 0.15$ | $2.89 \pm 0.17$ | $5.31 \pm 0.19$ | $3.91 \pm 0.32$ | $5.05^a \pm 0.15$ | $3.40^b \pm 0.25$ | $3.84^b \pm 0.38$ | $4.61^a \pm 0.32$ |
| WW 2010–11 | $5.73 \pm 1.12$ | $2.76 \pm 0.29$ | $4.48 \pm 0.48$ | $2.38 \pm 0.29$ | $5.10^a \pm 0.61$ | $2.57^b \pm 0.20$ | $4.25^a \pm 0.78$ | $3.43^a \pm 0.47$ |
| WW 2011–12 | $8.36 \pm 1.08$ | $4.55 \pm 0.52$ | $8.31 \pm 0.70$ | $4.40 \pm 0.35$ | $8.33^a \pm 0.60$ | $4.48^b \pm 0.29$ | $6.45^a \pm 0.91$ | $6.35^a \pm 0.82$ |
| Cover crop 2012–13 | $1.73 \pm 0.09$ | $1.88 \pm 0.17$ | $0.85 \pm 0.09$ | $0.96 \pm 0.10$ | $1.29^a \pm 0.18$ | $1.42^a \pm 0.19$ | $1.80^a \pm 0.09$ | $0.91^b \pm 0.06$ |
| Faba 2013 | $6.79 \pm 0.38$ | $3.38 \pm 0.12$ | $6.06 \pm 0.29$ | $2.86 \pm 0.40$ | $6.43^a \pm 0.26$ | $3.12^b \pm 0.22$ | $5.09^a \pm 0.67$ | $4.46^a \pm 0.65$ |
| WW 2013–14 | $9.34 \pm 1.45$ | $4.94 \pm 0.31$ | $9.61 \pm 0.20$ | $4.48 \pm 0.33$ | $9.48^a \pm 0.68$ | $4.71^b \pm 0.23$ | $7.14^a \pm 1.08$ | $7.04^a \pm 0.99$ |
| Cover crop 2014–15 | $1.88 \pm 0.13$ | $1.92 \pm 0.13$ | $2.58 \pm 0.12$ | $2.47 \pm 0.14$ | $2.23^a \pm 0.16$ | $2.20^a \pm 0.14$ | $1.90^b \pm 0.09$ | $2.52^a \pm 0.09$ |
| Maize 2015 | $10.07 \pm 0.90$ | $3.31 \pm 0.32$ | $8.46 \pm 0.40$ | $3.38 \pm 0.48$ | $9.27^a \pm 0.55$ | $3.34^b \pm 0.27$ | $6.69^a \pm 1.35$ | $5.92^a \pm 1.00$ |
| Global rate | $56.86 \pm 1.98$ | $27.16 \pm 0.82$ | $54.44 \pm 0.57$ | $26.51 \pm 1.00$ | $55.65^a \pm 1.06$ | $26.84^b \pm 0.611$ | $42.01^a \pm 5.7$ | $40.48^a \pm 5.3$ |

**Notes.**
WW, winter wheat; CT, conventional tillage; RT, reduced tillage; IN, incorporation of crop residue; OUT, exportation of crop residue.

management treatment, the mixed effects model was used with the depths as a fixed factor and plots and dates as random effects. A student's $T$-test was used to test for each treatment whether the soil factors of the last sampling year and the first sampling year were significantly different.

## RESULTS

### Management of crop residues

After seven crop rotations, the total amount of crop residue returned to the soil was on average 52% higher (i.e., +28.8 t/ha) for IN plots (55.7 t/ha) compared to OUT plots (26.8 t/ha) (Table 4 and Table S2 for ANOVA summary). This was correlated with an increase in the amount of nutrients in the residues (Tables S3 and S4) that were further restored to the soil.

While the stock (expressed in [kg/ha]; Table S3) of nutrients returned were greater in IN plots, the OUT plots were characterised by greater content (expressed in [g/kg]; Table S5) of N (three years out of five) and P (two years out of five), due to a larger proportion of chaff in the remaining residues. The trend was the opposite for K (two years out of five) (Tables S5 and S6).

**Table 5  Effect of residue treatment on the NPK stock of crop residues [kg/ha, ±standard deviation] restituted to the soil per type of crop residue source.** Wheat values came from the mean of three years of experiments in winter wheat. For each type of nutrient and crop, treatments with different letters are significantly different (ANOVA, $p$-value $< 0.05$).

| Nutrient | Crop | Residue treatment | |
|---|---|---|---|
| | | IN | OUT |
| N [kg/ha] | Wheat | $39.6^a \pm 14.3$ | $20.6^b \pm 6.7$ |
| | Faba bean | $63.9^a \pm 14.7$ | $43.5^b \pm 10.2$ |
| | Maize | $60.2^a \pm 13.8$ | $22.3^b \pm 5.4$ |
| P [kg/ha] | Wheat | $5.7^a \pm 2.5$ | $3.0^b \pm 1.3$ |
| | Faba bean | $8.6^a \pm 3.4$ | $4.4^b \pm 1.5$ |
| | Maize | $7.9^a \pm 0.8$ | $3.0^b \pm 0.8$ |
| K [kg/ha] | Wheat | $29.5^a \pm 21.5$ | $10.3^b \pm 5.6$ |
| | Faba bean | $42.2^a \pm 12.1$ | $13.8^b \pm 3.5$ |
| | Maize | $95.6^a \pm 27.9$ | $26.9^b \pm 4.3$ |

**Notes.**
IN, incorporation of crop residue;  OUT,  exportation of crop residue.

The tillage treatment had no significant impact on the total amount of crop residues or on the stock of nutrient (Table 4 and Tables S2–S6).

Table 5 puts the emphasis on the different amounts of each nutrient (NPK) returned by the different crops through their residues: e.g., while maize and faba bean brought higher N than wheat, maize residues alone provided the highest quantity of K to the soil.

## Soil results
### Total organic carbon
The initial TOC content, over the entire arable depth (0–30 cm), was 11.7 g/kg. After 8 years, the different treatments (tillage or residue management) did not have a significant effect on the overall TOC content evolution (lmer : on tillage $F = 1.62, p = 0.22$ ; on residue treatments $F = 0.58, p = 0.46$), although significant differences were observed during some specific years (Fig. 1). The last measurement, in spring 2016, showed higher TOC under RT-IN compared to CT-OUT and RT-OUT.

We observed a clear stratification of the TOC content between the different soil depths (0–10, 10–20 and 20–30 cm depths) in the reduced tillage treatments (lmer $F = 236.55$, $p < 2.2 e^{-16}$; Fig. 1B–1D). More specifically, the TOC content increased over time in the 0–10 cm soil profile and, after 3.5 years, was systematically higher in RT than in CT (with the exception of autumn 2014). At that depth, higher TOC contents were observed in RT-IN than in RT-OUT. In the 10–20 cm soil layer, there were no differences between CT and RT. In the 20–30 cm soil layer, RT resulted in lower TOC content than CT from the third year of the trial onwards.

### Soil nutrients
*Nitrate.*  Averaged over the soil profile and over time, we did not observe differences in nitrate content between the different treatments, although some temporary differences were visible (Fig. 2). The levels of nitrate within the 0–90 cm soil layer were highly variable
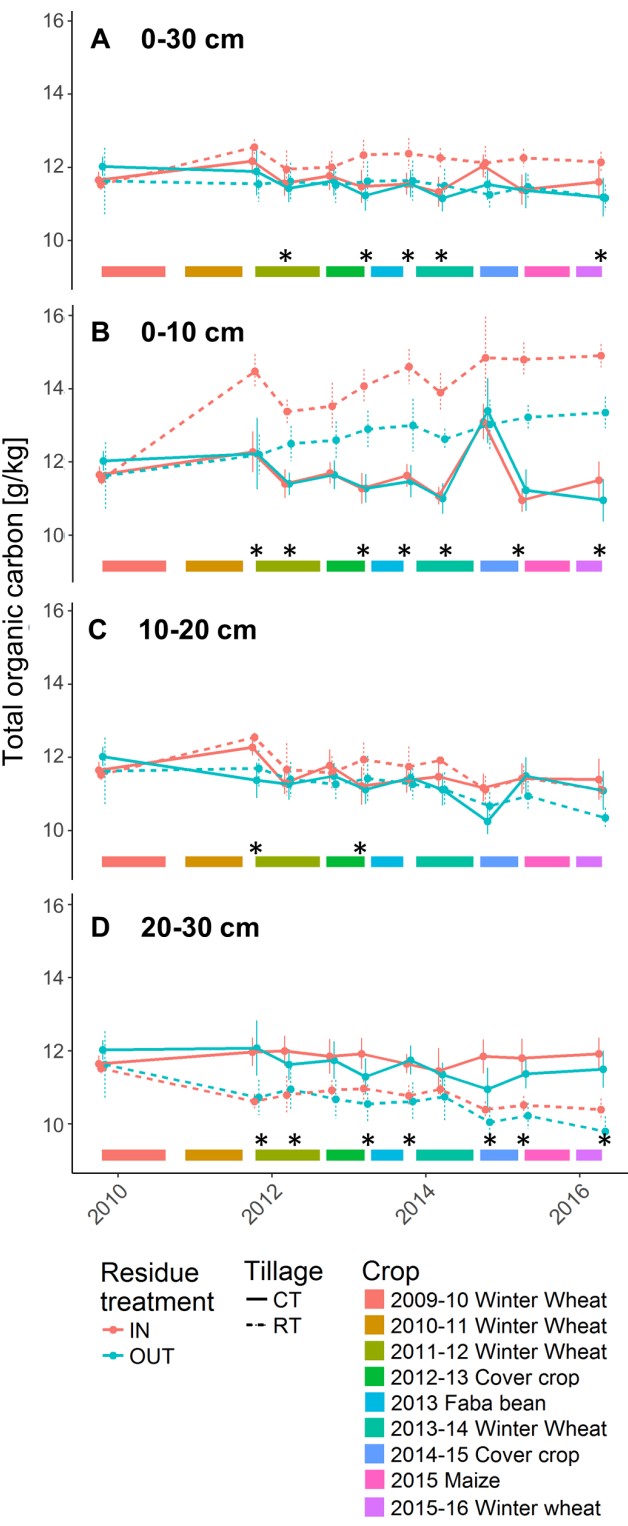

**Figure 1** **Total organic carbon (g/kg) present in the (A) 0–30 cm, (B) 0–10 cm, (C) 10–20 cm, (D) 20–30 cm soil layer.** Bottom ribbons represent the period covered by crops. Error bars depict the standard error between plots. Stars represent significant differences (ANOVA, *p*-value < 0.05) between crop residue management strategies per sampling date.

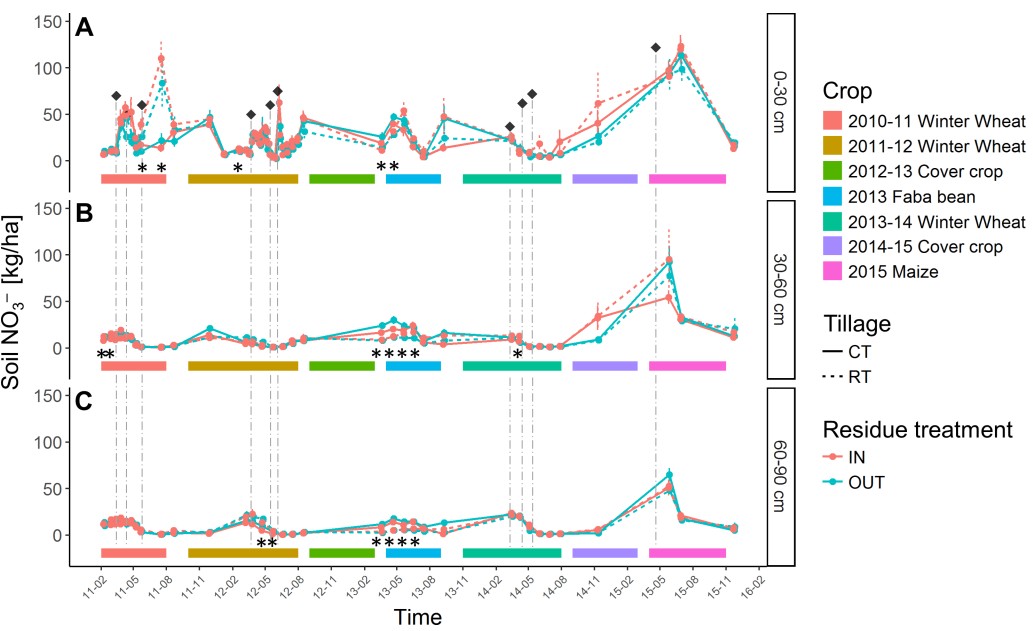

**Figure 2** **Nitrates (N- NO$_3^-$) (kg/ha) present in the (A) 0–30 cm, (B) 30–60 cm, (C) 60–90 cm depth layers of soil.** Bottom ribbons represent the period covered by crops. Error bars depict the standard error between plots. Vertical dot lines represent the date of crop fertilization. Black diamonds represent the quantity of nitrates added to the crop as mineral fertilization. Stars represent significant differences (ANOVA, *p*-value < 0.05) between crop residue management strategies per sampling date.

over time and dominated by external inputs of fertilizer (black dots on Fig. 2). Residue fate had no impact on the NO$_3^-$ stock in any of the soil layers under any crop.

*Phosphorus.* Over the course of the experiment, P content in soil significantly decreased in the 0–30 cm soil layer for all crop residue treatments, due to the absence of fertilisation during the course of the experiment (Fig. S4). In addition, there was a stratification of P under RT (RT-OUT from autumn 2012 onwards (lmer $F = 84.58$, $p < 2.2\ e^{-16}$) and RT-IN from spring 2014 onwards (lmer $F = 34.06$, $p = 3.6\ e^{-12}$)). In the top layer (0–10 cm), these treatments showed a higher decrease in P content than deeper in the profile. Our data do not suggest any significant impact of crop residue on P stocks in the soil (lmer $F = 0.05$, $p = 0.83$).

*Potassium.* As for P, K content in soil decreased from the beginning of the trial, as no K fertilisation was applied (Fig. S4). No particular effects of the different treatments were observed on the total amount of K (lmer $F = 0.06$, $p = 0.81$), except for the last 2 years of the experiment (2015–2016), where IN plots showed a higher K content. Also for K a stratification was visible (lmer $F = 97.91$, $p < 2.2\ e^{-16}$) with decreasing concentrations from top to bottom.

**Table 6** **Descriptive statistics for germination rate [%, ±standard error] for the crops grown during the study period 2009–2015.** For each crop, treatments means with different letters are significantly different (ANOVA, $p$-value $< 0.05$).

| | Interaction between fixed factors | | | | No interaction between factors | | | |
| | Crop residue management | | | | Residue fate | | Tillage type | |
| | CT-IN | CT-OUT | RT-IN | RT-OUT | IN | OUT | CT | RT |
|---|---|---|---|---|---|---|---|---|
| WW 2009–10 | $65.8 \pm 5.6$ | $63.4 \pm 3.1$ | $61.7 \pm 5.9$ | $66.8 \pm 4.9$ | $63.7^a \pm 3.8$ | $65.1^a \pm 2.8$ | $64.6^a \pm 3.0$ | $64.2^a \pm 3.7$ |
| WW 2010–11 | $60.5 \pm 0.9$ | $62.9 \pm 1.1$ | $49.1 \pm 1.5$ | $51.7 \pm 3.1$ | $54.8^a \pm 2.3$ | $57.3^a \pm 2.6$ | $\mathbf{61.7^a \pm 0.8}$ | $\mathbf{50.4^b \pm 1.7}$ |
| WW 2011–12 | $69.3 \pm 1.1$ | $69.0 \pm 4.6$ | $65.0 \pm 2.5$ | $71.0 \pm 2.4$ | $67.1^a \pm 1.5$ | $70.0^a \pm 2.4$ | $\mathbf{69.1^a \pm 2.2}$ | $\mathbf{68.0^a \pm 2.0}$ |
| Faba 2013 | $72.3 \pm 2.1$ | $74.3 \pm 0.9$ | $50.5 \pm 6.3$ | $55.5 \pm 2.9$ | $61.4^a \pm 5.1$ | $64.9^a \pm 3.8$ | $\mathbf{73.3^a \pm 1.1}$ | $\mathbf{53.0^b \pm 3.3}$ |
| WW 2013–14 | $\mathbf{72.0^a \pm 1.8}$ | $\mathbf{65.4^a \pm 3.0}$ | $\mathbf{53.7^b \pm 5.3}$ | $\mathbf{62.7^{ab} \pm 1.8}$ | $62.8 \pm 4.3$ | $64.0 \pm 1.7$ | $68.7 \pm 2.0$ | $58.2 \pm 3.1$ |
| Maize 2015 | $78.6 \pm 0.8$ | $78.2 \pm 1.9$ | $75.1 \pm 1.8$ | $77.1 \pm 0.9$ | $76.9^a \pm 1.1$ | $77.6^a \pm 1.0$ | $78.4^a \pm 0.9$ | $76.1^a \pm 1.0$ |
| Global rate | $\mathbf{69.7^a \pm 1.3}$ | $\mathbf{68.9^a \pm 1.1}$ | $\mathbf{59.2^c \pm 2.6}$ | $\mathbf{64.1^b \pm 1.6}$ | $64.4 \pm 2.4$ | $66.5 \pm 1.3$ | $69.3 \pm 0.8$ | $61.6 \pm 1.7$ |

**Notes.**

WW, winter wheat; CT, conventional tillage; RT, reduced tillage; IN, incorporation of crop residue; OUT, exportation of crop residues, Global rate is the average of all germination rates of all crops considered for a specific treatment, normalized to the sowing density.

## Crop results
### Germination rate

The presence or absence of extra residues did not affect germination rate, except for winter wheat in 2013–14 where we observed a lower germination rate in the RT-IN treatment. Tillage type, however, did result in differences in germination rate. Table 6 (and ANOVA summary in Table S7) shows that three crops (out of six) had a higher germination rate in CT as compared to RT (winter wheat (2010–11), faba bean (2013) and winter wheat (2013–14)).

In Table 6, we see that residue fate and tillage type had an interacting effect on the global relative germination rate which is the average of all germination rates of all crops considered for a specific treatment, normalised to the sowing density. The germination rate in RT-IN was significantly lower than RT-OUT, itself lower than CT-IN and CT-OUT.

### Dynamic crop growth

When looking at the dynamics of crop growth, except for two sampling dates on ears' growth and two sampling dates on shoot's growth, no interactions were found between tillage and residue management treatment. Observations and results of the ANOVA and SNK analysis are reported in the supplementary material separately for ears and shoot growth (Tables S8–S11). The cumulated total produced biomass is presented in Fig. S5 (ANOVA summary in Table S12) under graphical representation. It was therefore decided to analyse the impacts of tillage and residue management individually.

Differences between residues management treatments (IN *vs.* OUT) were observed for crop development. Incorporation of residues (IN plots) negatively impacted the dynamic
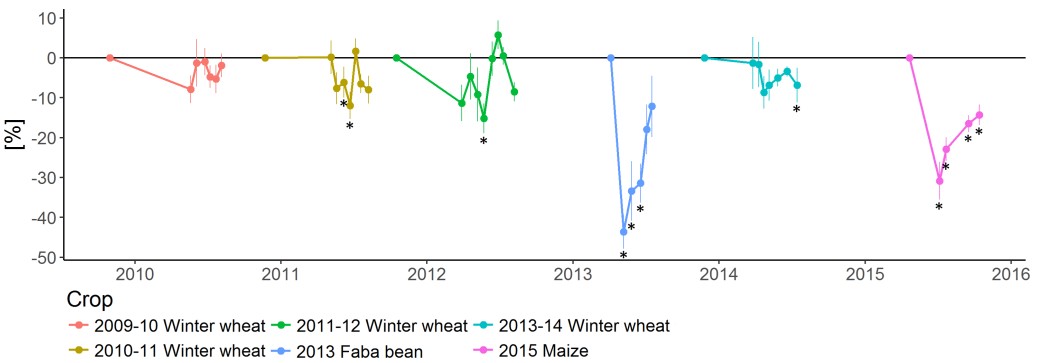

**Figure 3** **Relative crop biomass differences (%) between conventional tillage (CT) and reduced tillage (RT), relative to CT.** Error bars depict the standard error between plots. Stars represent significant differences (ANOVA, $p$-value $< 0.05$) between crop residue management strategies per sampling date.

of ears and shoots during the first two crop seasons (winter wheat seasons 2009–10 and 2010–11). These differences appeared in both cases after the ear emergence stage. These were more pronounced for the shoot than the ear biomasses. In 2010, with weather conditions close to historical means, lower biomass under IN treatments decreased over the crop growth period, but no negative impact on yields were observed when residues were returned to the soil. In 2011, characterised by a spring drought, the exportation of residue (OUT) was favourable to the shoot development and to the early growth of ears. However, for this season, the final yield was not impacted by the residue management treatment (IN *vs.* OUT), as detailed in the next section. Finally, one could also notice that at the end of the 2014 crop season, while no differences were observed all along the season, a statistical difference was reported on the last sampling date in favour of IN plots. For all other sampling dates, no statistical differences were reported.

Reduced tillage mostly negatively impacted shoot crop development of winter wheat (2010–11), faba bean (2013) and maize (2015) (Tables S8 and S9). This was probably due, in part, to the lower germination rate in RT plots for the winter wheat crop (2010–11)and faba bean (2013) (cfr 'Crop yield and quality'). When computing the differences between tillage treatments (Fig. 3 and Table S12 for ANOVA summary), it was observed that the gap between treatments tended to decrease for the different crops as they developed through the season. For faba bean no differences were finally observed between the final biomass (Fig. 3), the pods (Tables S10 and S11) and the seed yield (Table 7—cfr 'Crop yield and quality'), while statistical differences between RT and CT remained for maize shoot, cobs and grain yield cultivated in 2015 (Fig. 3, Tables S10 and S11, and Table 7—cfr 'Link between germination, development and yield').

### Crop yield and quality

Yearly grain yields were in general not influenced by residue fate or soil tillage (Table 7 and Table S13 for ANOVA summary), except for a negative effect for reduced tillage for winter wheat cultivated in 2010–11 (−9%) and maize in 2015 (−4%) and residue incorporation

**Table 7  Descriptive statistics of yield dry weight [%, ±standard error] for the crops grown during the study period 2009–2015.** For each crop, treatments means with different letters are significantly different (ANOVA, $p$-value $< 0.05$).

| | Interaction between fixed factors | | | | No interaction between factors | | | |
| --- | --- | --- | --- | --- | --- | --- | --- | --- |
| | Crop residue management | | | | Residue fate | | Tillage type | |
| | CT-IN | CT-OUT | RT-IN | RT-OUT | IN | OUT | CT | RT |
| WW 2009–10 | 8.64 ± 0.09 | 8.56 ± 0.12 | 8.01 ± 0.47 | 8.32 ± 0.12 | 8.33[a] ± 0.25 | 8.44[a] ± 0.09 | 8.60[a] ± 0.07 | 8.17[a] ± 0.23 |
| WW 2010–11 | 6.95 ± 0.13 | 7.59 ± 0.12 | 6.23 ± 0.16 | 7.05 ± 0.12 | **6.59[a] ± 0.17** | **7.32[b] ± 0.13** | **7.27[a] ± 0.15** | **6.64[b] ± 0.18** |
| WW 2011–12 | 6.56 ± 0.11 | 6.53 ± 0.07 | 6.73 ± 0.38 | 6.83 ± 0.09 | 6.65[a] ± 0.19 | 6.68[a] ± 0.08 | 6.54[a] ± 0.06 | 6.78[a] ± 0.18 |
| Faba 2013 | 4.01 ± 0.20 | 4.02 ± 0.20 | 3.98 ± 0.10 | 3.81 ± 0.17 | 3.99[a] ± 0.10 | 3.91[a] ± 0.13 | 4.01[a] ± 0.13 | 3.90[a] ± 0.10 |
| WW 2013–14 | 7.67 ± 0.07 | 7.74 ± 0.18 | 7.63 ± 0.07 | 7.56 ± 0.14 | 7.65[a] ± 0.05 | 7.65[a] ± 0.11 | 7.71[a] ± 0.09 | 7.60[a] ± 0.07 |
| Maize 2015 | 10.66 ± 0.09 | 10.66 ± 0.42 | 10.40 ± 0.15 | 9.99 ± 0.33 | 10.53[a] ± 0.09 | 10.32[a] ± 0.28 | **10.66[a] ± 0.20** | **10.19[b] ± 0.19** |
| Global rate | 44.49 ± 0.03 | 45.10 ± 0.62 | 42.98 ± 0.45 | 43.57 ± 0.45 | 43.73[a] ± 0.35 | 44.34[a] ± 0.46 | **44.79[a] ± 0.31** | **43.27[b] ± 0.32** |

**Notes.**

WW, winter wheat; CT, conventional tillage; RT, reduced tillage; IN, incorporation of crop residue; OUT, exportation of crop residues.

for winter wheat in 2010–11 ($-10\%$, year characterised by a spring drought, Figs. S2 and S3).

The cumulative grain yield since 2010 was significantly lower under reduced compared to conventional tillage ($-3.4\%$, Table 7). No effect of residue fate was observed.

There was no significant effect of the treatments on NPK content of the harvested grain or seeds (Table S14), except for marginally higher P content in winter wheat (in 2010–11) grains in IN plots ($P$-value: 0.03) and a slightly lower K content under conventional tillage in 2 years out of 5 (Winter wheat in 2011–12, $p$-value: 0.01; Maize in 2015, $p$-value: 0.03).

### *Link between germination, development and yield*

No correlation was seen between germination rate and shoot dry weight except for faba bean (2013) and a slight effect for winter wheat (2010–11) (Fig. 4A). Grain yield was only positively correlated to germination rate for winter wheat (2010–11) (Fig. 4B). Similarly, grain yield was slightly correlated to shoot dry weight for that year only (Fig. 4C).

## Integrated approach between crop production and soil chemistry

A principal component analysis was performed to study the relationship between soil and plant parameters (Fig. 5). Plant and crop residue data are the sums (for residue biomass, shoot biomass, yield and NPK stocks) or means (for germination rate and harvest index (HI)) of the all crop data (the six crops grown between 2009 and 2015). We used soil data from the last spring measurement in 2016.

The two principal components allowed to explain respectively 35% and 25% of the variance. Treatments were easily differentiated by the two principal components (Fig. 5A).

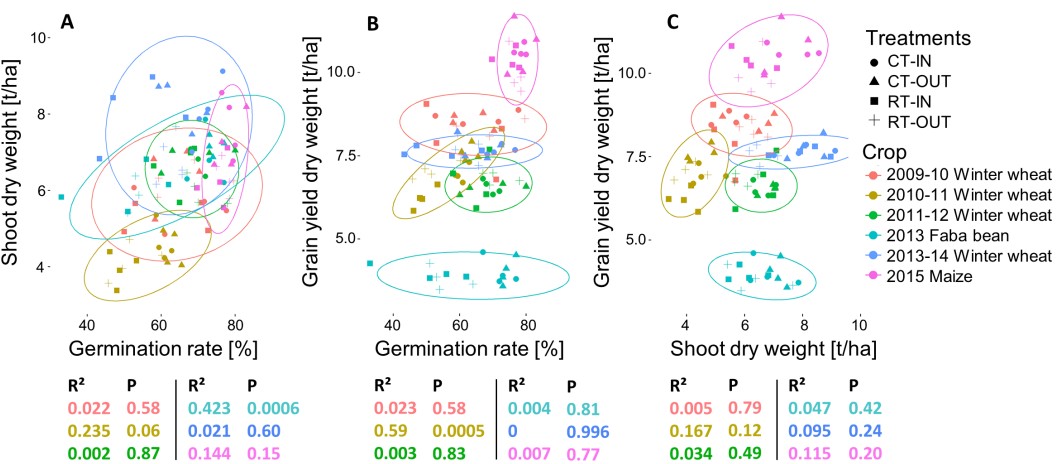

**Figure 4** **(A) Total shoot dry weight (t/ha) versus germination rate (%). (B) Grain yield versus (t/ha) germination rate (%). (C) Grain yield (t/ha) versus total shoot dry weight (t/ha).** Ellipses encompass 95% of the distribution. Below each graph, colour-related *r*-squared values and *p*-values are given for that crop.

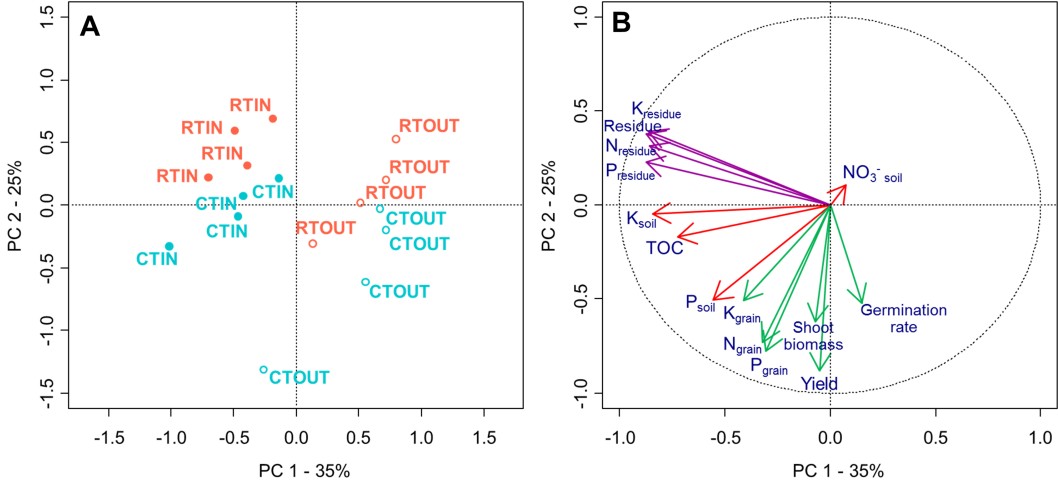

**Figure 5** **(A) Unconstrained ordination analysis. (B) Correlations between soil and plant variables.** (A) Each dot represents an individual plot (from 1 to 16). Red is reduced tillage and blue is conventional tillage. Filled dots are for plots with incorporation of crop residues and empty dots are plots with exportation of crop residues. (B) Residue data are represented by purple arrows, Soil parameters are in red arrows and crop parameters in green. ($NO_3^-$, nitrates in the 0–90 cm soil layer; TOC, total organic carbon; K, potassium; P, phosphorus; N, nitrogen). TOC, K and P in soil are value of the 0–30 cm final measurement. Plant and crop residue data are the accumulation (Residue biomass, shoot biomass, yield, NPK stock) or mean (germination rate, harvest index (HI)) of the all crop data. Soil data are the data measured in spring 2016 (last measurement).

It appeared that crop productivity (yield, shoot and total biomass) and quality ($K_{grain}$, $P_{grain}$ and $N_{grain}$) were favoured by conventional tillage, as illustrated by the discrimination of treatments along the second component ($Y$-Axis) of the PCA. Similarly, it seemed that the soil parameters (TOC, $K_{soil}$ and to a lower extent $P_{soil}$) were more positively influenced by the crop residue retention (Fig. 5B), as illustrated by the discrimination of treatments along the first component ($X$-Axis) of the PCA.

## DISCUSSION

### Effect of tillage and crop residue treatments on crop production

Overall, tillage influenced crop production more strongly than import or export of residues. The strongest effect was seen in terms of germination rate and was even stronger for residue incorporation treatments. RT resulted consistently in lower germination rates as also shown by *Brennan et al. (2014)*. Germination rate is strongly affected by seedbed soil moisture, soil structure and contact around the seed and soil temperature (*Guérif et al., 2001*). It is acknowledged that crop residues can be a physical obstacle to crop emergence and a source of phytotoxicity for crop seedlings (*Morris et al., 2010*). Moreover, the presence of crop residues around seeds can impede adequate seed-to-soil contact needed for good crop emergence by increasing the macroporosity which is known to decrease the degree of contact (*Brown et al., 1996*). Nevertheless, the differences due to germination rate have the tendency to disappear at later growth stages if no climatic extremes occur, since the plants generally compensate a lower density with a better growth under favourable growth conditions as also observed by *Dam et al. (2005)*.

Several studies report higher crop production levels under CT under temperate climate, when compared to RT (*Brennan et al., 2014*; *Pittelkow et al., 2014*). However, the difference between both systems remained small in the presented experiment, but confirmed, among else, by the PCA analysis. Our results went in the direction of the conclusion of *Van den Putte et al. (2010)* study of conservation agriculture in Europe showing a yield decline of 4.5% in RT systems as compared to conventional systems. When confronting the crop results to the meteorological conditions (Fig. 3, Table 7 and Figs. S2 and S3) we believe that the CT production systems might be less sensitive to inter-annual fluctuations of climatic conditions over different years compared to the RT systems. Also, *Brennan et al. (2014)* highlighted that the residue fate was less important than the tillage type for crop performance. Residue fate has a stronger effect on crop production under drier climates and water limited conditions (*Linden, Clapp & Dowdy, 2000*; *Pittelkow et al., 2014*).

The differences observed for germination rates or during crop growth seem to have little influence on crop yield. A similar lack of correlation between shoot biomass and grain yield, as observed most strongly for faba bean and wheat (2014) in our study, has previously been observed for legume crops (*Araújo & Teixeira, 2008*). For winter wheat, it is likely that the ability to produce more tillers at lower densities explains the recovery from lower germination rates, as reported in the literature (*Whaley et al., 2000*; *Gooding, Pinyosinwat & Ellis, 2002*). Moreover, it is known that the flag leaf and ears are the main photosynthetic organs contributing to grain filling (*Sanchez-Bragado et al., 2014*) which

is an additional explanation why the entire shoot biomass was not correlated to yield, especially during years with climate conditions close to the historical means (i.e., winter 2009–10, 2011–12 and 2013–14). Winter wheat (2010–11) and maize (2015) were the only crops with observable differences between treatments at the end of crop development. We hypothesize that the spring drought during winter wheat development in 2010–11 impeded its ability to recover its potential yield as in the other years.

Except for the winter wheat 2010–11, our results did not show an increase in N grain stock. This observation is in agreement with *Brennan et al. (2014)*, but opposed to the results reported by *Malhi et al. (2011)* or *Soon & Lupwayi (2012)*. The absence of crop residue treatment effects on P grain content could be also due to the poor P content of crop residues. Regarding the small differences observed in grain K content, *Zörb, Senbayram & Peiter (2014)* mentioned that K content in grain is not correlated to K supply and grain have relatively low K contents.

## Effect of tillage and crop residue treatments on soil chemistry

Over the seven years of this experiment, the effects of crop residue management on soil fertility parameters showed few statistical differences in the early time of the trial. However, the results were slowly magnifying, up to the point where differences became more systematically significant, with clear stratification occurring over the different soil layer of the ploughing depth. Furthermore, our results (PCA analysis) confirmed a clear link between soil TOC, K content, and to a lower extent P content, with residues management treatment (IN *vs.* OUT).

Even though the literature shows that residue incorporation could have a positive effect on the stock of SOC (*Chenu et al., 2014*; *Autret et al., 2016*; *Merante et al., 2017*), we only observed small effects on the TOC content. It should be noted that we cultivated wheat for four out of seven years and therefore our residues contain a large proportion of straw, which has already been shown to be inefficient to increase TOC (*Lemke et al., 2010*; *Poeplau et al., 2015*). Just like previous studies (*Angers et al., 1997*; *Dolan et al., 2006*; *Gadermaier et al., 2012*; *Dimassi Bassem, 2013*; *Riley, 2014*; *Dikgwatlhe et al., 2014*), we have shown that reduced tillage provoked a stratification of TOC. The absence of differences between TOC(CT-IN) and TOC(CT-OUT) can be explained by a dilution effect (accumulation of organic matter in the top that is mixed through ploughing) as well as by a potentially faster degradation rate, as reported by *Lal et al. (2004)*.

The absence of any overall effect of residue treatment on the nitrate content was likely due to a combination of factors. Firstly, the proportionally high amount of mineral N applied as fertiliser—which respects the common practice in Belgium—reduced the effect of N returned by the residues. Secondly, the straw incorporation effect might have had a short-term impact on soil nitrate (*Van Den Bossche et al.*) rather than long term impact (*Brennan et al., 2014*). Such a lack of impact was also reported by *Stenberg et al. (1999)*.

The absence of residue treatment effects on P content in soil can be explained by the low P content of crop residues. (*Damon et al., 2014*) have shown that P availability is only increased for large amounts of residues with high P content. A P content threshold of 2–3 mg/g of residue is generally considered as the limit below which no impact

should be expected. Under this value, immobilisation by microbial biomass occurs and P mineralization is hampered (*Damon et al., 2014*). In our study, P content was 0.9 mg/g for wheat and maize residues and 1.6 mg/g for faba bean crop, which means that we were consistently and considerably below this theoretical threshold for P mineralization.

The decrease in K content was due to the lack of fertilisation during the trial (no dedicated K fertilisation or manure application was intentionally made between 2008 and 2016) but this decrease is slight. Compared to P content, K content of residues was much higher. Furthermore, as the mineralisation process is not involved, K is released in soil solution as soon as the plant cells were dead (*Schvartz, Decroux & Muller, 2005*). These combined effects probably explain the slighter slope observed on the dynamics of K. The stratification observed in P and K content with reduced tillage was also reported by *Riley (2014)*.

## CONCLUSION

When looking within the available choices among the soil and crop management techniques, crop production remains one of the most important drivers of farmer's decisions. However, the impact of residue management and tillage treatment on crop production, and also on soil fertility, are known to be highly dependent on the local pedo-climatic conditions, and the literature usually focus on one of the aspects (soil or crop) and barely on multiple aspects at the same time.

This study aimed at analysing the impacts of crop residue management techniques on different soil and crop parameters together, as interacting components of the agro-ecosystem, and as a response to the local pedo-climatic conditions. We found out that, at the annual scale, crop production was generally not significantly impacted by the different residue management strategies. However, over the duration of the trial, while no effect of residue fate was reported (IN *vs.* OUT), the cumulative grain yield was found to be significantly lower (−3.4%) under reduced tillage (RT) compared to conventional tillage (CT).

In this seven-year experiment, small but gradually increasing differences between the different crop residue management strategies were observed. After a few years, the TOC content in the soil was higher only where the residues were incorporated and the tillage reduced. Overall, a stratification of organic matter and nutrients was observed under reduced tillage, i.e., for TOC, P and K. Crops grown on reduced tillage plots had a lower germination rate in some years, but in two years out of three crops overcame this germination gap through compensation mechanisms and finally yields were statistically equivalent.

Soil processes in general, and carbon dynamics in particular, are slow processes. Our study reflects a system currently in transition, which will likely continue to evolve over the next decade. Therefore, it will be of uttermost importance to continue the monitoring of this experimental site in order to understand the long-term impact of residue management on crop performance and soil quality and health.

Finally, when choosing among soil and crop management techniques, (among which the fate of residues and the intensity of tillage that were analysed in this study are few examples),

other factors than the sole crop productivity should be included in the farmer's decision process, such as fuel consumption, required working hours, greenhouse gas emissions (*Lognoul et al., 2017*); analysis conducted on the same experiment), soil fauna (*Degrune, 2017*), long-term soil quality and health.

## ACKNOWLEDGEMENTS

A special thanks to the entire technical team of the experimental farm for their helping hands in the field. We also want to thank the CRA-W for their support. We also thank Yves Brostaux for the statistical advices.

### Funding

This work was funded by the AgricultureIsLife research platform—Gembloux Agro-Bio Tech—University of Liège. The CRA-W also supported part of the measurements. The funders had no role in study design, data collection and analysis, decision to publish, or preparation of the manuscript.

### Grant Disclosures

The following grant information was disclosed by the authors:
AgricultureIsLife research platform—Gembloux Agro-Bio Tech—University of Liège.

### Competing Interests

The authors declare there are no competing interests.

### Author Contributions

- Marie-Pierre Hiel performed the experiments, analyzed the data, prepared figures and/or tables, authored or reviewed drafts of the paper, approved the final draft.
- Sophie Barbieux and Claire Olivier performed the experiments.
- Jérôme Pierreux performed the experiments, authored or reviewed drafts of the paper.
- Guillaume Lobet authored or reviewed drafts of the paper.
- Christian Roisin contributed reagents/materials/analysis tools.
- Sarah Garré authored or reviewed drafts of the paper, approved the final draft.
- Gilles Colinet and Bernard Bodson conceived and designed the experiments, contributed reagents/materials/analysis tools, authored or reviewed drafts of the paper, approved the final draft.
- Benjamin Dumont analyzed the data, authored or reviewed drafts of the paper, approved the final draft.

### Data Availability

Hiel, Marie-Pierre (2017): Crop residue management : crop ans soil raw data (2010-2016). figshare. Fileset. https://doi.org/10.6084/m9.figshare.5442289.v1.

## Supplemental Information

Supplemental information for this article can be found online at http://dx.doi.org/10.7717/peerj.4836#supplemental-information.

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
