# Peer review of "Impact of crop residue management on crop production and soil chemistry after seven years of crop rotation in temperate climate, loamy soils"

_PeerJ, doi:10.7717/peerj.4836_

## Round 0.1 · original submission · Major Revisions

· Academic Editor

Major Revisions

Dear Marie-Pierre,

Thank you for your submission to PeerJ. Please address the reviewer comments as fully as possible.

Sincerely,

Aan Ribeiro-Barros

·

Basic reporting

The manuscript is generally well written in good English. However, the abstract should be more explicit, more clear/precise and could be more concise (see specific comments). In the introduction section, please define specific research questions and/or hypotheses, which can be tested. It becomes clear why you do this long-term experiment, but the hypotheses could be stated more clearly. In addition, the results part is poorly structured and therefore makes some of the discussion sections hard to understand. In the discussion, you should not only compare your values to literature, but also discuss why in your particular experiment you observed These specific effects and what you can conclude from this. At the moment, the conclusions are weak and not always supported by your findings. Additionally, you argue that the duration of the trial might have been too short to find significant effects, but you don't say what you would expect in the long-term.

Experimental design

The latin square design of the long-term trial is clear and makes sense for the purpose of the experiment. However, as mentioned above, the research questions are kept very general and there are no testable hypotheses. In addition, there is some lack of details in the methods and a basic soil characterisation (initial pH, soil organic carbon, texture, CEC, N, P, K concentrations...) is missing. This should be added.

Validity of the findings

Even though, the statistics are well described in the methods section, the statistical results are not reported adequately. 2-way ANOVA would be appropriate with the main factors tillage (RT, CT) and residue management (IN, OUT) and the ANOVA table should be reported (e.g. in the supplementary information). Then, treatment means should be compared with an appropriate post hoc test such as Tukey HSD. If you average over treatment levels, this should be clearly stated and argued why it was done.

Generally, I agree, that your long-term trial contributes to the understanding of yield developments and nutrient dynamics under different tillage and residue management and that the results are highly relevant for sustainable agricultural practises.

Additional comments

One specific comment here: It would be good if you reported the residue and grain N, P, K concentrations in mg/kg and not only the imports and exports in kg/ha.

Reviewer 2 ·

Basic reporting

The manuscript “Impact of crop residue management on crop production and soil fertility after seven years of crop rotation in a temperate climate and loamy soil” evaluates the effect of residue management and tillage on the crop production and soil properties. It is praiseworthy that authors have done extensive data collection over several years period. However, I regret that this manuscript cannot be published as such due to very inconsistent and poor presentation. Specifically, the material and method section is not clear and the results presented (along with Tables and Figures) are not provided with sufficient statistical analyses.
I have following suggestion to authors to improve this manuscript:
Abstract:
L30: Authors talk about six crops, but only three are given; winter wheat, faba bean and maize, what are other three crops?
L38: Please first give abbreviation for nutrient, and then use abbreviated form later in the text.
L45: incomplete sentence

Introduction
L60: “However, the actual impact of straw incorporation is often discussed (Poeplau et al. 2015)”. Please write what does this study suggest; positive, negative or either both impacts of the straw incorporation.

Experimental design

The method section is not sufficiently described.
L108-109 states “ It has to be noted that stubble and chaff are always left on the fields, even if the rest of the residue is exported”. However, Line 133-134 states that residue (OUT: Stubble and Chaff). These two statements create confusion on residue management strategies. Were the stubble and chaff left on the field or were they exported? Please clarify the treatment description.
L117-118: What equipment was used for sowing?
L136 : what is composite grain sample. For example, it is the ears or cobs or grain of maize, or pods or seeds of faba bean and similarly the case for wheat?
L137: “ We also collected the residues present on 1 m2 per plot…..does this area represent the statistical unit. Residue NPK content can vary depending on which part used for sampling, for instance, the grain chaff or the straw? Replication in the residue NPK determination is the must.
In crop sampling and Analysis:
Describe in detail the method of harvesting for the individual crop, as well their residue harvesting?
L146: how was soil sampling conducted? Was it done using Auger or Giddings probe? How were soil samples composited?
L147: what was the mass of soil sample used for soil organic carbon determination? Was it done in the fresh or air dried sample?
L152-157: Similar to SOC determination how was sampling conducted for nitrate? Were Augers used? How was nitrate content determined, was the sample air dried or fresh?
L162-163: why statistical test not reported when the interactions were not significant, were there no main effect as well?
Statistical Analyses:
How the crop growth parameters; germination, yield... and soil properties over different years test? Were there any repeated measure analysis?

Validity of the findings

Results need to be supported by strong statistical analyses for the validity.
Result:
L184-185: three out of six crops…where and which are other three crops?
L189: what was the sowing density for different crops; wasn't it different for different crops? How can sowing density for different crops such as faba bean, wheat and maize be averaged?
L192-196: where are the results presented? Tables/Figures?
L200-206 and Figure 1: were there any statistical differences, please give p values.
L211 and throughout the text: please present important result in figures or tables, not in Appendix.
Table 7: Please do not use common after each number, if it is the decimal point then use so.
Table 3 and 4: What is global rate?
In tables general:
Write down if the comparisons are made within the row or column.
Please present standard error instead of stand deviation, also in case of figures.
In Figure 4: what do single and double stars represent?

Discussion
In several instances, authors suggested that similar findings are already established in the literature such as
“L293-295- A similar lack of correlation between shoot biomass and grain yield…..has previously been observed……
L319-321-Just like previous studies………we showed reduced tillage provoked ……
L330-331-Such a lack of impact was also reported…..”
Readers are therefore left wondering what new does this study suggests besides already established facts in literature. More discussion on the results is required.

Annotated reviews are not available for download in order to protect the identity of reviewers who chose to remain anonymous.

Reviewer 3 ·

Basic reporting

This work focuses on the impacts of crop residue management (returned to soil vs exported) and type of tillage (reduced tillage depth vs conventional ploughing) on crop production and soil chemical properties. The effects of these management strategies were evaluated on a 7-year field-experiment under a temperate climate and a loamy soil. This paper presents interesting results showing some trends in differentiation for soil properties on the short-term, which potentially suggest more important changes on the long term.

Overall, the manuscript is clear and well structured. The background and the research question (lines 88-93) are clearly introduced and defined, the field experiment and methods are well suited to answer the questioning and the results are nicely discussed.

As a non-native English speaker, I do not feel well qualified to evaluate the language used.
Specific comments are detailed in my comments to the authors.

Experimental design

The field experimental design detailed in part 2.2 (lines 105-127) makes, in my opinion, the strength of this paper and provides a solid foundation to develop interesting and robust researches on the short-term as exposed in this paper as well as on the long term (perspectives of this work).

The sampling strategy (10 soil subsamples forming representative composite samples at 3 different depths) allows to take into account spatial variability and to analyze the effects of crop residue managements and tillage types at different soil depths. This last point is particularly important as many studies only focused on a thin soil surface layer and have drawn some conclusions (e.g. on soil C storage under no tillage) without considering a potential stratification effect on the whole soil profile.

The laboratory and statistical analyses are globally well described and sound.
I was just wondering if additional data related to i) nitrate measurements and ii) soil bulk densities were available, which could maybe provide complementary results interesting for this paper:

i) In this study, “a stratification of organic matter and nutrients was observed under reduced tillage… for carbon, phosphorus and potassium” (lines 360-361), but no differences in nitrate contents were observed between the different treatments (lines 249-250). However, according to the methods, it should be noted that the effects on nitrate contents were measured on different soil layers and depths (0-30, 30-60 and 60-90 cm) compared to carbon and the other nutrients, which were measured for 0-10, 10-20 and 20-30 cm. Are data available for nitrate contents in these latter soil layers? Due to soil stratification under reduced tillage on 0-30 cm, an effect on soil nitrate could maybe be captured if this layer would be more in depth analyzed, even though the great mobility of nitrate and the important amounts of fertilizer used here could maybe prevent the detection of a subtle effect.

ii) It could be also interesting to have measurements of soil bulk densities in order to evaluate the effects of reduced tillage on soil compaction and to analyze the evolution of soil organic carbon stocks for the different treatments.

In addition, it should be explained how were calculated nutrient contents (in kg/ha).

Validity of the findings

The conclusions of this work are well supported by the results obtained, with a robust set of data followed by a rigorous interpretation. As stated by the authors, the results obtained here are relevant for the specific pedoclimatic condition of this study, due to the importance of climate and soil parameters on the investigated processes.

Additional comments

Specific comments:

Title:
I would suggest to add “tillage” in the title. Fertility could be replaced by another term (e.g. chemistry…) as fertility may refer to many other aspects.

Abstract:
Maybe not very stylish to close the abstract with “…” (line 45). Furthermore, the last sentence is perhaps not essential and a little bit long. It could be suggested to finish the previous sentence (line 43) by something like “… whether more consistent changes in the whole agro-ecosystem functioning…” to include more concisely the idea developed in the last sentence.

Introduction:
Line 64: “Hiel et al. (2016)” is not in the reference list.
Line 77: I would suggest to replace “say something about” by “assess”
Line 78: “show” -> “showed”

Methods:
Lines 133-135 refer to residue analyses and line 136 to grain samples, and then, again to residues, line 137. Maybe better to group the sentences related to residues together.
Part 2.4. As explained above, it should be explained here how were calculated nutrient contents (in kg/ha).

Results:
In line 204 are presented results for fababean yields from Table 5. To be consistent within the different parts, this should be included in the next part 3.1.3 relative to crop yields.
Line 223: “crop residue in the soil” -> “crop residue returned to soil”
Line 224: “this was correlated with an increase in the amount of nutrient”, maybe interesting to add statistical criteria for this correlation.
Lines 225-227: I was not able to find the results presented here in the tables.
Lines 259-260: “these treatments showed a slower decrease in P content than deeper in the profile”. This is not clear, according to the figure, there is a higher decrease of P in RT treatments in the upper layer. Please clarify. Legend of the figure in Appendix D: “Phosphore” -> “Phosphorus”

Discussion:
Line 311: “soil fertility”: same comment than for the title, maybe use a more specific term.
Line 314 and Appendix E: I would suggest to comment this result and this figure in the result part.
Lines 320-321: Maybe also add in the references the work of Dimassi et al. (2013 or 2014 in AGEE) in which a similar stratification of soil organic carbon was observed following tillage conversion in a long term experiment.

Conclusion:
Lines 348-349: “was not dramatically impacted…” I would suggest to add “on the short term”.

---

## Round 0.2 · Minor Revisions

· Academic Editor

Minor Revisions

Dear Dr Benjamin Dumont

Thank you for re-submitting the revised version of your MS. All questions raised by the reviewers have been properly addressed and the MS has increased considerably. However, I still found some minor details that can further improve the MS. These are basically related to the improvement of the text rather than the data. Please see the minor comments in the attached version, and send me the revised version, highlighting only the new changes proposed or explaining why they were not included.

After receiving this final version, I will immediately accept the MS.

Than you once again for considering PeerJ to publish your work,

Ana Ribeiro-Barros
Academic Editor, PeerJ

---

## Round 0.3 · accepted · Accept

· Academic Editor

Accept

Dear Dr Marie-Pierre Hiel and Dr Benjamin Dumont,

Thank you for sending the revised version of your manuscript including the proposed minor revisions. It is my pleasure to inform you that the paper is now accepted for publication.

Thank you for considering Peer J to publish your work,

Sincerely,
Ana I. Ribeiro-Barros

#